# Poly[3-ethyl-1-vinyl-imidazolium] diethyl phosphate/Pebax^®^ 1657 Composite Membranes and Their Gas Separation Performance

**DOI:** 10.3390/membranes10090224

**Published:** 2020-09-08

**Authors:** Irene R. Mazzei, Daria Nikolaeva, Alessio Fuoco, Sandrine Loïs, Sébastien Fantini, Marcello Monteleone, Elisa Esposito, Saeed Jamali Ashtiani, Marek Lanč, Ondřej Vopička, Karel Friess, Ivo F. J. Vankelecom, Johannes Carolus Jansen

**Affiliations:** 1Institute on Membrane Technology (CNR-ITM), Via P. Bucci 17/C, 87036 Rende (CS), Italy; irene.r.mazzei@durham.ac.uk (I.R.M.); m.monteleone@itm.cnr.it (M.M.); e.esposito@itm.cnr.it (E.E.); johannescarolus.jansen@cnr.it (J.C.J.); 2Membrane Technology Group (MTG), cMACS, Faculty Bio-science Engineering, Celestijnenlaan 200F, 3001 Leuven, Belgium; ivo.vankelecom@kuleuven.be; 3SOLVIONIC, Site Bioparc 195, route D’Espagne, BP1169, 31036 Toulouse CEDEX 1, France; sandrinelois@orange.fr (S.L.); sfantini@solvionic.com (S.F.); 4Department of Physical Chemistry, University of Chemistry and Technology, Technická 5, 166 28 Prague, Czech Republic; Saeed.Jamali.Ashtiani@vscht.cz (S.J.A.); Marek.Lanc@vscht.cz (M.L.); Ondrej.Vopicka@vscht.cz (O.V.); karel.friess@vscht.cz (K.F.)

**Keywords:** poly(ionic liquids), Pebax^®^ 1657, gas transport, membrane separation, composite materials

## Abstract

Poly(ionic liquid)s are an innovative class of materials with promising properties in gas separation processes that can be used to boost the neat polymer performances. Nevertheless, some of their properties such as stability and mechanical strength have to be improved to render them suitable as materials for industrial applications. This work explored, on the one hand, the possibility to improve gas transport and separation properties of the block copolymer Pebax^®^ 1657 by blending it with poly[3-ethyl-1-vinyl-imidazolium] diethyl phosphate (PEVI-DEP). On the other hand, Pebax^®^ 1657 served as a support for the PIL and provided mechanical resistance to the samples. Pebax^®^ 1657/PEVI-DEP composite membranes containing 20, 40, and 60 wt.% of PEVI-DEP were cast from solutions of the right proportion of the two polymers in a water/ethanol mixture. The PEVI-DEP content affected both the morphology of the dense membranes and gas transport through the membranes. These changes were revealed by scanning electron microscopy (SEM), time-lag, and gravimetric sorption measurements. Pebax^®^ 1657 and PEVI-DEP showed similar affinity towards CO_2_, and its uptake or solubility was not influenced by the amount of PIL in the membrane. Therefore, the addition of the PIL did not lead to improvements in the separation of CO_2_ from other gases. Importantly, PEVI-DEP (40 wt.%) incorporation affected and improved permeability and selectivity by more than 50% especially for the separation of light gases, e.g., H_2_/CH_4_ and H_2_/CO_2_, but higher PEVI-DEP concentrations lead to a decline in the transport properties.

## 1. Introduction

Although the purpose of CO_2_ capture encourages innovative research in the design of new gas separation technologies, their industrial implementation remains a major challenge that needs to be urgently addressed [1]. Membrane-based technologies represent a modular, affordable, and environmental-friendly alternative to several industrial techniques for separation and purification of CO_2_ from gas mixtures, such as natural gas purification or CO_2_ recovery from flue gas produced by combustion of fossil fuels [2,3,4]. The improvement of membrane performance in these industrial separation processes requires the constant search for novel materials with enhanced performance by tailoring permeability and selectivity properties [5]. 

In the last decade, ionic liquids (ILs) emerged as promising materials for improved CO_2_ separation properties of membrane-based systems [6]. ILs possess advantageous characteristics, such as negligible vapor pressures, broad thermal stability and chemical flexibility. Their ionic nature allows the modification of their physico-chemical properties through various cations and/or anions combinations [7]. ILs are commonly used in the fabrication of supported liquid membranes (SLMs), wherein the desired IL is trapped inside a porous polymeric matrix by capillary forces [8]. However, the membranes produced accordingly are often prone to IL leaching from the pores [9]. This issue can be solved by the exploitation of a polymerizable group on the IL cation, to create self-standing materials through a polymeric backbone. Such IL-based polymers are commonly addressed as polymeric or polymerized ionic liquids (PILs) and represent a class of polyelectrolytes [10,11,12]. The use of PILs as membranes for gas separation was initially introduced by Noble’s group. They predicted the performance of 1-R-3-methylimidazolium (Rmim)-based membranes for CO_2_/N_2_ and CO_2_/CH_4_ separations [13]. 

Several studies highlighted the strong dependence of the performance of ILs and PILs in gas separation processes, focusing on high CO_2_ selectivity and the nature of their cations [5,12,14,15,16,17,18,19,20]. Ammonium and imidazolium-based PILs, in particular, demonstrated an improved affinity with CO_2_ and advantageous CO_2_/CH_4_ separation performance [10,21,22]. Other studies focused on the role of counter-anions and observed significant effects as well [18,23]. Bis(trifluoromethylsulfonyl) imide ([Tf_2_N]¯) anion is often used in the preparation of ILs. However, the relatively high cost of the starting material Li[Tf_2_N] promotes the search for alternatives [24,25,26]. While a variety of anions is available on the market (e.g., Ac^−^, BF_4_^−^, PF_6_^−^), the phosphate anion represents a promising alternative based on its availability, cost, and chemical stability. Although some commercial ILs with alkyl-phosphate anions were used in gas separation applications [16], to the best of our knowledge, there are no reports on the alkyl-phosphate counter-anions application in PILs-based gas separation membranes.

Despite the advanced separation properties, PILs still face some challenges concerning their functionalization, chemical stability, and mechanical strength. The latter can be considerably improved by blending PILs with a robust polymer. There are only a few reports about the preparation of PIL/polymer composite membranes, which are mainly investigated for electrochemical devices and wastewater purification [27]. Pebax^®^ 1657 is an excellent polymer for the fabrication of composite membranes with PILs due to its rubbery nature, the presence of polar, CO_2_-phylic, polyethylene oxide (PEO) segments in its structure, and its remarkable mechanical properties [28,29,30,31,32]. Despite already showing good performance in the separation of CO_2_ from non-polar species, numerous attempts have been made to further improve the separation properties of Pebax^®^ 1657, by combining the polymer with ILs [33,34]. Recently, a composite Pebax^®^ 1657/IL membrane containing the 1-n-alkyl-3-methylimidazolium cation showed considerable improvement in CO_2_ permeability [35]. However, the IL in liquid physical state may still be prone to leaching. Thus, the combinations of robust polymers and tailor-made PILs in composite membranes for gas separation may benefit both classes of materials. The confirmed high affinity of imidazolium-based PILs towards CO_2_ could be combined with resistant and well-defined polymers, to prepare novel families of membranes with enhanced separation properties.

The present work investigates the role of poly[3-ethyl-1-vinyl-imidazolium] diethyl phosphate (PEVI-DEP) in gas separation performance of Pebax^®^ 1657/PEVI-DEP composite membranes, prepared by blending and solvent casting. The PEVI-DEP/Pebax^®^ 1657 gas transport parameters were studied for the first time using sorption experiments and time-lag measurements for single and mixed gas permeation. Correlation with different PEVI-DEP contents and morphological aspects was expected to reveal the influence of PIL on the gas transport mechanism.

## 2. Materials and Methods

### 2.1. Materials

Poly[3-ethyl-1-vinyl-imidazolium] diethyl phosphate and Pebax^®^ 1657 (pellets) were kindly provided by SOLVIONIC (Toulouse, France) and Arkema (Milan, Italy), respectively (Figure 1). Pebax^®^ 1657 was used as received, whilst PEVI-DEP was dried in an oven at 100 °C for 20 h prior further use. Demineralized water and ethanol used for the fabrication of dense membranes were supplied by VWR (Milan, Italy). The gases used in permeation tests (nitrogen, oxygen, methane, helium, hydrogen, and carbon dioxide, minimum purity of 99.9995%) were supplied by Sapio (Monza, Italy). 

### 2.2. Membrane Preparation 

Dense composite membranes were solvent cast from Pebax^®^ 1657/PEVI-DEP blends. 8 wt.% of Pebax^®^ 1657 in EtOH/H_2_O (70/30 wt.%) was stirred for 24 h at room temperature and subsequently heated at 80 °C under reflux for 2 h. Pebax^®^ 1657/PEVI-DEP solutions with PEVI-DEP content of 20 wt.%, 40 wt.% and 60 wt.% were prepared by adding PEVI-DEP to the solution while still hot and stirred till homogeneous. The solutions were cast onto Petri dishes in a controlled environment at 25 ± 1 °C and 20 ± 1 % relative humidity and left to solidify for ca. 120 h. The solid membranes were removed from the Petri dishes and dried in a vacuum oven overnight prior to further handling.

### 2.3. Scanning Electron Microscopy (SEM)

SEM images of the surface of the fabricated composite membranes were acquired using a Phenom ProX desktop SEM (Phenom-World B.V., Eindhoven, Netherlands) in backscattered electron mode. The samples were placed on a stub using adhesive carbon tape and analyzed without sputter coating at an acceleration voltage of 10 kV. 

### 2.4. X-ray Diffraction (XRD)

The X-ray diffraction (XRD) analysis was performed at room temperature using the Bruker D8 (Bruker, Berlin, BB, Germany) Discoverer θ-θ diffractometer with para-focusing Bragg-Brentano geometry and CuKα radiation (λ = 0.15418 nm, U = 40 kV, I = 40 mA). Data were scanned over the angular range 5–80° (2θ) with a step size of 0.05° (2θ), 3791 steps, and a total effective time of 1943s. Data processing was performed with the software package DIFFRAC.SUITE™.

### 2.5. Fourier-Transformed Infrared Spectroscopy (FT–IR)

Fourier-transformed infrared spectroscopy (FT–IR) measurements were conducted on a Nicolet iS50 FTIR Spectrometer ((Thermo Scientific, Boston, MA, USA). The membrane samples were fixed onto the attenuated total reflection (ATR) diamond crystal surface, and a deuterated triglycine sulfate detector was used for the measurements in the range of 500–3500 cm^−1^, with a resolution of 0.09 cm^−1^, for all samples.

### 2.6. Time-Lag

Single gas time-lag experiments were carried out on a fixed volume/pressure increase instrument designed by Helmholz Zentrum Geesthacht and constructed by Elektro and Elektronik Service Reuter (Geesthacht, Germany) on circular membrane samples. A description of the experimental apparatus can be found elsewhere [36]. The membranes were degassed in the test cell using a turbo molecular pump before being tested. The feed gas pressure was set to 1 bar for all gases. The permeate pressure was measured up to a maximum of 13.3 mbar with a resolution of 0.0001 mbar. The gases were always tested in the following order: H_2_, He, O_2_, N_2_, CH_4_, and CO_2_. Permeability (*P)* is reported in Barrer (1 Barrer = 10^−^^10^ cm^3^ (STP) cm·cm^−^^2^·s^−^^1^·cm Hg^−^^1^) and defined according to Equation (1) as the product of the diffusion coefficient *D* (m^2^·s^−^^1^) and the solubility *S* (cm^3^ (STP) cm^−^^3^·bar^−^^1^). The diffusion coefficient was calculated from the permeation time-lag Θ (s) and knowing the membrane thickness *l* (measured using a micrometer screw gauge), according to Equation (2).
(1)P=D · S
(2)$$\Theta =\;\frac{l^{2}}{6\;D}$$

The ratio of the permeability over the diffusion coefficient provides an indirect measurement of the gas solubility [36,37,38]. The selectivity (*α*, dimensionless) between two gas species was calculated as the ratio of the parameters measured for each species. 

### 2.7. Gravimetric Sorption

Gravimetric sorption experiments were performed on an in-house designed apparatus equipped with a McBain balance for N_2_, CH_4_, and CO_2_, using a method described elsewhere [39,40,41,42]. The sample (approximately 0.15 g) placed in a thick-walled glass tube was evacuated (<10^−3^ mbar) prior to each measurement by a rotary oil pump (Leybold Trivac D4B) until a constant weight was achieved. The gas was then flushed in the chamber and the quartz spiral elongation was recorded by a charge-coupled device (křemenná spirála, Brno, SM, Czech Republic), leading to the gravimetric sorption determination. Sorption experiments were carried out in the range of gas pressures from 1 bar to 10 bar, at room temperature. 

Sorption isotherms in a dense, homogeneous matrix generally follow Henry’s law, which describes a linear correlation between the gas uptake and the pressure applied: (3)$$c=k_{D}\;p$$
where *c* (cm^3^ (STP) cm^−3^) is the volumetric concentration of gas in the sample, *p* (bar) is the pressure, and *k_D_* (cm^3^ (STP) cm^−3^·bar^−1^) is Henry’s sorption constant. When micropores are dispersed in a dense polymeric matrix, two kinds of sorption processes contribute to the total amount of adsorbate in the material [43,44]. In this case, the sorption isotherm is described by the dual-mode model [45,46] (4). According to the dual-mode theory, sorption in the dense phase follows Henry’s law whilst the uptake of gas into the voids is described by Langmuir’s model.
(4)$$c=k_{D}\;p+\frac{c_{H}bp}{1+bp}$$

In Equation (4), *c_H_* (cm^3^ (STP) cm^−3^) and *b* (bar^−1^) derive from Langmuir’s model and take into account, respectively, the effects of a gradual saturation of the defects in the matrix of the membrane and the affinity between gas and polymer. 

### 2.8. Mixed Gas Analysis

Mixed gas permeation measurements were performed on a custom-made variable volume/constant pressure instrument. The permeate, retentate, and feed compositions are measured using a mass-spectrometric residual gas analyzer (HPR-20 QIC, Hiden Analytical, Warrington, UK). Measurements were performed in a cross-flow cell, at high feed flow rates and relatively high flow rate of Argon used as the sweeping gas. This allows us to avoid polarization phenomena by having a low stage cut and gives negligible partial pressure in the permeate. Details on the instrument, measurement, and calibration procedures were described in earlier works [47,48]. 

## 3. Results and Discussion

### 3.1. Morphology and Bulk Properties

Pictures of the membranes and SEM images of their surface are shown in Figure 2. Clear effects of the addition of the PIL in Pebax^®^ 1657 were (i) a change of color from colorless of neat Pebax^®^ 1657 to brown, and (ii) a decrease of the mechanical strength. Specifically, membranes became more fragile but maintained their soft nature (i.e., samples were not brittle). At high PEVI-DEP concentrations the SEM images show evident needle-like features in the polymer blend, which we partially ascribe to PEVI-DEP domains, indicating low miscibility between the PIL and Pebax^®^ 1657. We believe that PEVI-DEP clusters cause an increase in the distance between Pebax^®^ 1657 polymeric chains, which affects significantly the mechanical properties of the membranes. This is due to the radically different nature of the two materials: Pebax^®^ 1657 exhibits good mechanical strength due to the presence of hard PA segments, whereas the character of the PIL was rather viscous at room temperature and did not exhibit any mechanical resistance.

XRD patterns of the membranes with different amounts of PEVI-DEP are shown in Figure 3a. The semicrystalline character of Pebax^®^ 1657 is demonstrated by a broad amorphous halo (12.5–22.6°) and a dominant peak around 24° (2θ) representing the crystalline segments of the PA domains. The presence of PEVI-DEP in the Pebax^®^ 1657 matrix caused a shift in the PA peak and a corresponding increase in the d-spacing from 24.29° and 3.664 Å for neat Pebax^®^ 1657, to 23.79° and 3.739 Å for the Pebax^®^ 1657/60 wt.% PEVI-DEP sample, confirming our previous hypothesis. This shift can be attributed to the intercalation of the charges of the cations in the PIL to influence the Pebax^®^ 1657 polymer chain backbone [50] and is in agreement with earlier observations that low-molar-mass [BMIM][CF_3_SO_3_] interferes with the crystalline phase of Pebax^®^ 1657 [35]. Simultaneously, it indicates that the addition of PIL did not affect the inter-chain hydrogen bonding between PA segments in Pebax, since the hindering of their formation generally leads to the growth of the amorphous region [51]. Furthermore, the Pebax^®^ 1657/PEVI-DEP membranes spectra contain another peak at 19.23° originating from the presence of crystalline ionic liquid-like domains.

The observed peaks at 2881 cm^−^^1^ and 1557 cm^−^^1^ in all the FTIR spectra (Figure 3b) represent the stretching vibration of asymmetric and symmetric primary amines, and secondary amines groups from the PA segments of Pebax^®^ 1657. Their identical position for the neat Pebax^®^ 1657 and Pebax^®^ 1657/PEVI-DEP samples implies that there is no chemical change in the membrane matrix after the addition of PIL, and thus supporting the physical blending nature of ionic liquid within the Pebax matrix in the microphase domains and confirming the low miscibility. Two observed minor blue-peak-shifts related to the increasing content of PIL in the membranes, from 1644 cm^−^^1^ and 1117 cm^−^^1^ to 1630 cm^−^^1^ and 1085 cm^−^^1^, can be attributed to the and C–O stretching vibrations, and the decrease in the peak-intensity with increasing PIL content reveals a weak tendency of the carbonyl group to form hydrogen bonding with the PIL.

### 3.2. Gas Transport Properties

Figure 4 and Appendix A (Table A1) demonstrate the gas transport parameters of Pebax^®^ 1657/PEVI-DEP composite membranes. Overall, the trends of gas permeability and diffusivity exhibit a rise with increasing PEVI-DEP content (Figure 4a,c). More specifically, the permeabilities reach maximum values for all gases in the membrane prepared with 40 wt.% of PEVI-DEP. However, CO_2_ and CH_4_ deviate from this trend, as their permeability remains almost constant or slightly decreases with increasing PEVI-DEP content. The transport reflects a combination of the effects of PEVI-DEP on the morphology and microstructure of the membranes and the effect on the individual phases. The weak increase in permeability for the smaller gases, together with a slight increase in size-selectivity, is due to a combination of effects caused by the addition of PEVI-DEP: reduced crystallinity of samples and an increase of their amorphous fraction leading to an increase of the overall permeability, since the crystalline phase is impermeable.

Simultaneously, the decrease in crystallinity did not lead to the expected (tangible) decrease in overall stiffness, which means that the effect is apparently compensated by a stiffening of the amorphous phase, either by the presence of PEV-DEP itself or by the presence of a larger amount of the hard polyamide chain segments in the amorphous phase. Since the amorphous phase is the only phase where gas transport takes place, for smaller gases the increase in the amorphous phase dominates and the sample becomes more permeable, but for larger gases stiffening compensates the effect of the increase of the amorphous fraction. At high PEVI-DEP content, the stiffening of the amorphous phase dominates, and the permeability and diffusivity of all gases decreases. The solubility values (Figure 4e) were not affected significantly by PEVI-DEP and remain almost constant. This observation contrasts with the original hypothesis stating that the incorporation of PIL into Pebax^®^ 1657/PEVI-DEP composite membranes should improve CO_2_ affinity, and this is most likely due to the already high sorption capacity of Pebax^®^ 1657. It should be noted that the time lag of He and H_2_ is near the lower measurement limit, causing a higher scatter in D and S for these two gases, which are therefore not included in the plots. 

### 3.3. Gas Sorption Properties

Sorption isotherms in Figure 5a,b show that all samples adsorbed CO_2_ in a slightly nonlinear fashion, which can be described satisfactorily by the dual-mode sorption model (Equation (4)). The sorption of CH_4_ and N_2_ was too close to the lower detection limit and could not be determined with sufficient accuracy. Therefore, the sorption isotherms of CH_4_ and N_2_ are not reported. The low CH_4_ and N_2_ uptakes in Pebax^®^ 1657/PEVI-DEP composites align well with other literature reports. The gas sorption follows the decreasing trend N_2_ ≤ CH_4_ << CO_2_ as reported for a series of poly(diallyldimethylammonium) (P[DADMA])-based PILs [52]. While the PIL concentration in the Pebax^®^ 1657/PEVI-DEP membranes has a relatively mild effect on the CO_2_ sorption capacity, the change in the isotherm shape is clearly distinguishable. Figure 5b compares sorption behavior of neat Pebax^®^ 1657 and neat PEVI-DEP. While the CO_2_ sorption isotherm of neat Pebax^®^ 1657 increases almost linearly with pressure, with a dominant role of Henry’s law, the sorption isotherm of neat PEVI-DEP exhibits concave shape attributed to dual-mode sorption. Dual-mode-like features are, therefore, transferred to Pebax^®^ 1657/PEVI-DEP composite membranes as well. The CO_2_ sorption isotherms (and data reported in Appendix A (Table A2)) confirm the earlier observation that the gas solubility is nearly independent of the PEVI-DEP content, especially at low concentration, similar to that of the permeability measurements.

### 3.4. Robeson Plots and Correlations

Robeson plots summarize the performance of Pebax^®^ 1657/PEVI-DEP composite membranes and put them in perspective for several industrial gas separation processes (i.e., CO_2_/CH_4_, CO_2_/N_2_, H_2_/CH_4_, O_2_/N_2_, and H_2_/CO_2_) (Figure 6). With increasing amounts of PEVI-DEP, the selectivity towards CO_2_ decreases, and the selectivity towards light gases increases. Previous reports demonstrate remarkable improvements of CO_2_ separation performance in Pebax^®^ 1657 thanks to the addition of ILs [33,34,35]; however, a drop of selectivity was observed for CO_2_/CH_4_ and CO_2_/N_2_ separations (Figure 6a,b) in Pebax^®^ 1657/PEVI-DEP composites. Therefore, the combination of the two materials did not meet the expectation of improving CO_2_ separation processes. In particular, the performances of the membrane containing 60 wt.% of PEVI-DEP were fully controlled by the PIL and were very similar to those of other PIL and PIL/IL based membranes [5,22,53,54,55,56]. However, especially in CO_2_/N_2_ separation, Pebax^®^ 1657/60 wt.% PEVI-DEP composite membrane is more selective than other PIL and PIL/IL membranes. These pieces of evidence indicate that CO_2_/CH_4_ and CO_2_/N_2_ separations are mostly favored by the presence of Pebax^®^ 1657. By contrast, H_2_/CH_4_ separation (Figure 6c) benefited from the addition of PEVI-DEP, showing an increase of selectivity and H_2_ permeability, which take Pebax^®^ 1657/PEVI-DEP samples closer to the performance of recently reported phosphonium-based PILs [55]. For H_2_/CO_2_ separation (Figure 6e) a trend that moves towards an inversion of selectivity was observed. This is due to simultaneous loss and gain of permeability for CO_2_ and H_2_, respectively. Moreover, the addition of PEVI-DEP substantially improved O_2_ and N_2_ permeabilities (for PEVI-DEP contents up to 40 wt.%), with a small gain in O_2_/N_2_ selectivity as well (Figure 6d). A common feature observed for O_2_/N_2_, H_2_/CH_4_ and H_2_/CO_2_ separations is a decrease of selectivity detected in the membrane prepared with 60 wt.% PEVI-DEP, compared to samples prepared with 40 wt.% and 20 wt.% of PIL. However, despite the modest increase in selectivity and permeability for some gas pairs, Pebax^®^ 1657/PEVI-DEP membranes are still rather far from recent Upper Bound limits [57,58].

The observed behavior, with a deviation from the trend for the sample with the highest PEVI-DEP content, may be due not only to the intrinsically different properties of the blended polymers but also to a change in the microstructure of the samples. At high PEVI-DEP content, the presence of crystal-like particles becomes more evident in the XRD patterns, and thus the sample heterogeneity. Although it is common to calculate the effective transport parameters using the Equations (1)–(3), a more correct evaluation would require the application of the Maxwell model or more sophisticated models for heterogeneous systems. Nevertheless, for practical application of the membranes, knowledge of the effective parameters is sufficient.

Figure 7 illustrates the correlations of transport parameters *D* and *S* with the effective diameter and the critical temperature of the gases, respectively [60]. The correlation between the logarithm of the diffusion coefficient and the square of the effective gas diameter defined by Teplyakov and Meares [60] reveals a linear decrease with increasing gas diameter. The slope of the line is a measure of the size-selectivity of the membranes. Besides some occasional scatter in individual points, there is no clear correlation between the PIL content and the size-selectivity. Interestingly, the diffusivity of CO_2_ is substantially lower than that of N_2_, especially at 20 wt.% and 40 wt.% PIL, suggesting that CO_2_ interacts with the PIL and is slightly slowed down. The slower diffusion of big molecules in the Pebax^®^ 1657 /PEVI-DEP membranes, probably due to specific non-covalent interactions [61], confirms a dominant role of diffusion-controlled transport in these materials. The plot of the solubility against the critical temperature of the gases (Figure 7b) also shows a nearly linear trend, with the highest values for CO_2_, and with little differences between neat Pebax^®^ 1657 and the composite membranes. The decrease in diffusivity and the increase in solubility with increasing size of the gas molecule is a typical feature for dense gas separation membranes [62]. 

Mixed gas permeation measurements were carried out across a total pressure range of 1–6 bar using a 35/65 vol.% of CO_2_/CH_4_ mixture for the two representative samples, namely Pebax^®^ 1657/20 wt.% PEVI-DEP and Pebax^®^ 1657/40 wt.% PEVI-DEP. Both membranes show very similar transport properties to those measured with pure gases in the time-lag setup (Figure 6a). The feed pressure barely affects the transport parameters, which are constant over the investigated range for both membrane compositions (Figure 8). This allows stable performance during industrial operations with variable pressure and composition, such as the biogas upgrading. 

## 4. Conclusions

Pebax^®^ 1657 composite membranes with PEVI-DEP as PIL additive were tested for gas sorption and separation properties. The gas separation measurements by time-lag method revealed a modest improvement of permeability compared to neat Pebax^®^ 1657 for H_2_, He, O_2_, and N_2_ and of selectivity in O_2_/N_2_, H_2_/CH_4_, and H_2_/CO_2_. This increase is mostly associated with better diffusion properties of small molecules in viscous PIL domains in the polymeric matrix. The best gas transport performance was observed for the membrane with 40 wt.% of PEVI-DEP in Pebax^®^ 1657. However, a higher PEVI-DEP content inverts the trend probably due to the change in the microstructure visible in the SEM images and XRD patterns. Sorption tests with CO_2_ showed a similar gas uptake for Pebax^®^ 1657 and PEVI-DEP. Despite not providing a higher affinity for CO_2_, and therefore an enhancement of CO_2_ separation performance, the addition of PEVI-DEP improves the efficiency of diffusion-controlled separations. This work demonstrated that the addition of specific amounts of PEVI-DEP to polymer/PIL composite membranes allows the fabrication of tailor-made dense membranes and improves processing properties of the polymer, using the viscous PIL as an additive in the polymeric matrix, but only up to a maximum of 40 wt.% of PEVI-DEP. Moreover, the membranes result stable at different feed pressures and under mixed gas permeation, proving to be reliable for real industrial operations. Further studies could explore the use of PILs with different structures (i.e., backbone and/or counter-anion) as additives to Pebax^®^ 1657 to modify and control its gas separation properties.

## Figures and Tables

**Figure 1 membranes-10-00224-f001:**
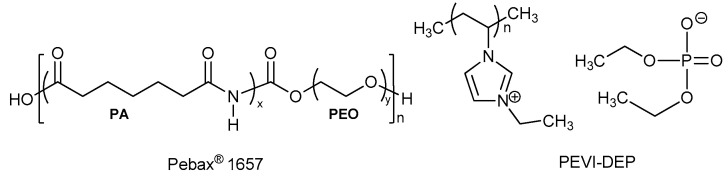
Structures of Pebax^®^ 1657 and PEVI-DEP (PA = polyamide, PEO = polyethylene oxide).

**Figure 2 membranes-10-00224-f002:**
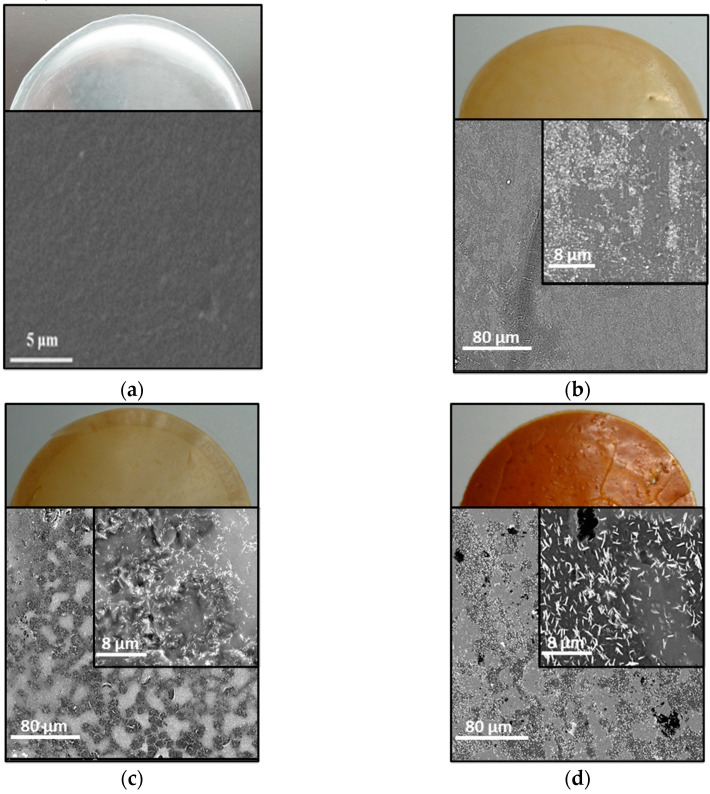
Photographs and SEM images of samples containing (**a**) 0 wt.%, (**b**) 20 wt.%, (**c**) 40 wt.%, and (**d**) 60 wt.% of PEVI-DEP in Pebax^®^ 1657. SEM images in (**b**–**d**) were acquired at magnifications of 1000× and 10,000×. The SEM image in (**a**) was adapted from Peng et al. [49].

**Figure 3 membranes-10-00224-f003:**
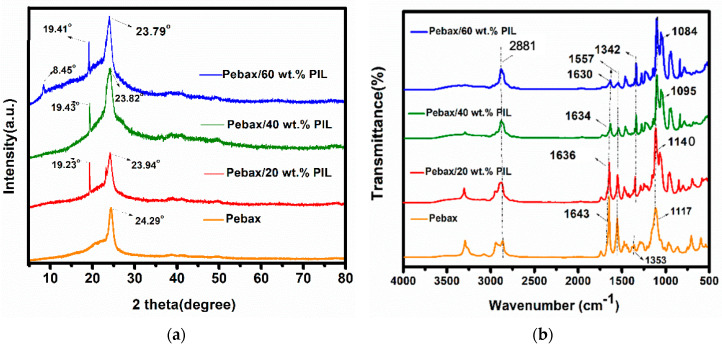
(**a**) The XRD patterns of the Pebax^®^ 1657 membranes blended with 0-60 wt.% of PEVI-DEP (**b**) The FTIR spectra of fabricated neat Pebax^®^ 1657 and Pebax^®^ 1657/PEVI-DEP membranes.

**Figure 4 membranes-10-00224-f004:**
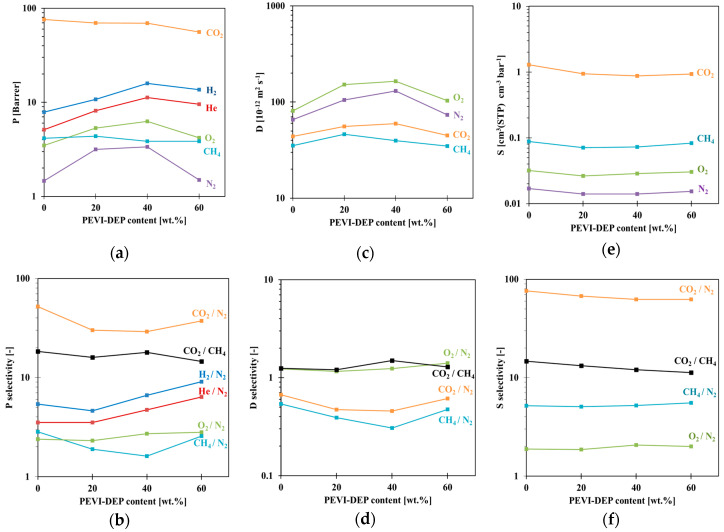
(**a**) Permeability, (**c**) diffusivity, and (**e**) solubility with relative selectivities (**b**,**d**,**f**) of neat Pebax^®^ 1657 and Pebax^®^ 1657/PEVI-DEP membranes.

**Figure 5 membranes-10-00224-f005:**
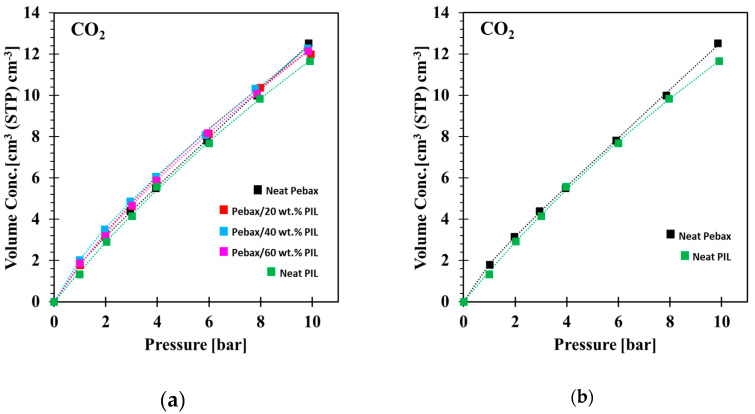
(**a**) Sorption isotherms of CO_2_ in Pebax^®^ 1657 membranes containing increasing amounts of PEVI-DEP and (**b**) comparison between neat Pebax^®^ 1657 and neat PEVI-DEP sorption isotherms.

**Figure 6 membranes-10-00224-f006:**
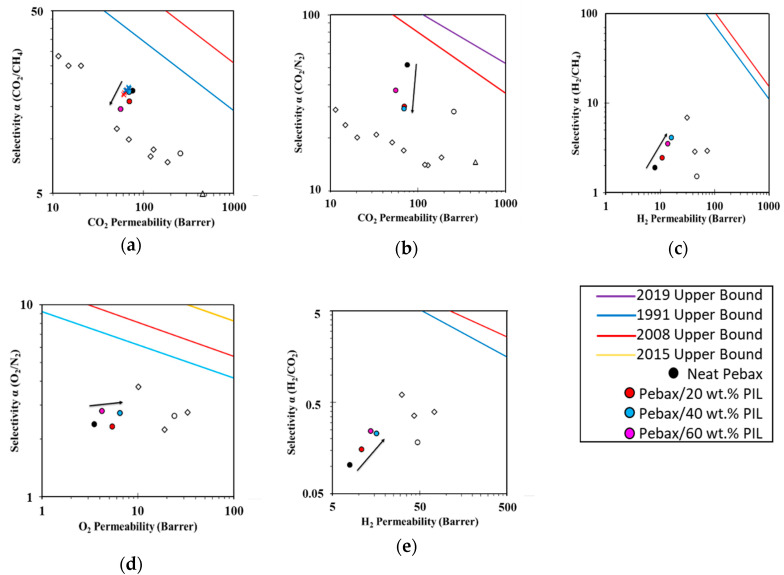
Robeson plots of neat Pebax^®^ 1657 and Pebax^®^ 1657/PEVI-DEP membranes analyzed in this work (filled circles) for (**a**) CO_2_/CH_4_, (**b**) CO_2_/N_2_, (**c**) H_2_/CH_4_, (**d**) O_2_/N_2_, and (**e**) H_2_/CO_2_ separations. Black arrows indicate the trend followed by the samples with increasing PEVI-DEP content. Data measured for the CO_2_/CH_4_ (35/65 vol.%) mixture at different feed pressures are indicated as “X” symbols with the same colour as the corresponding membrane. Other literature data for PILs [5,22,53,54,55,56] (empty diamonds), IL with the same anion as PEVI-DEP [59] (empty triangle) and Pebax^®^ 2533 [35] (empty circle) are reported for comparison.

**Figure 7 membranes-10-00224-f007:**
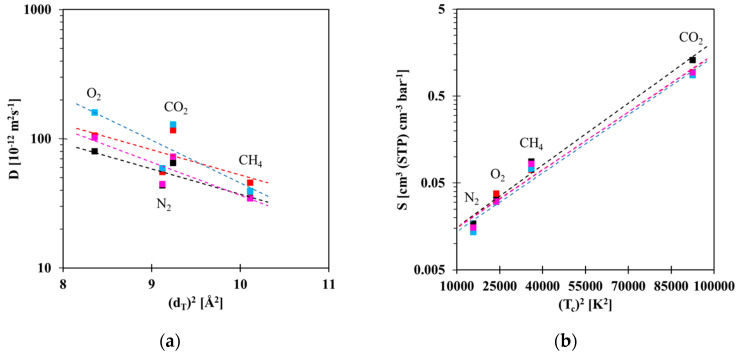
Correlations of (**a**) diffusion coefficient with Teplyakov’s diameters [60] and (**b**) solubility with critical temperatures of gas molecules. Linear trends are represented with dashed lines serve as a guide to the eye only.

**Figure 8 membranes-10-00224-f008:**
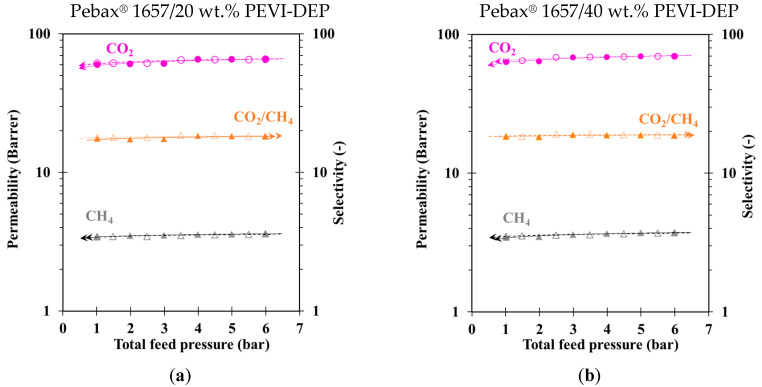
Pressure dependence of permselectivity of (**a**) Pebax^®^ 1657/20 wt.% PEVI-DEP and (**b**) Pebax^®^ 1657/40 wt.% PEVI-DEP with mixture CO_2_/CH_4_ (35/65 vol.%).

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
