# Peer review of "Poly[3-ethyl-1-vinyl-imidazolium] diethyl phosphate/Pebax^®^ 1657 Composite Membranes and Their Gas Separation Performance"

_membranes, 2020, doi:10.3390/membranes10090224_

Round 1
Reviewer 1 Report
The work under review discussed the formation of a mixed matrix system PEBAX/PIL. The authors describe how the insertion of the chosen PIL into PEBAX 1657 increased the permeability of all small gases with the exception of CO2. The authors suggest that the formation of a crystal-like microstructure at high IL contents. This might explain the improvement of both permeability and permeability selectivity of small gases vs. N2/CH4/CO2 by the formation of local defects at the interface between the rubbery polymer and PIL domains. To support this, it could be helpful to see the Wide Angle X-ray Diffraction spectra for all tested films.
CO2/CH4 and CO2/N2 permeability selectivity and CO2 permeability decreased with an increase of the PIL content. This might be due to the fact that the CO2 diffusion coefficient in the ILs domains is lower than in the PEBAX matrix (as discussed by the authors).
Overall the paper does not discuss a technology improvement on materials for membranes; instead, it is interesting to see how the addition of PIL to polymer films not always results in an improvement of CO2/CH4 and CO2/N2 gas separation properties. That is why I think the manuscript should be accepted for publication in the Membranes journal. This work is well written, and results are adequately discussed and referred to previously published relevant literature on the theme.
- That sais, as a minor addition to the paper content, I request that the authors include the WAXD spectra of all tested films, and discuss the effect of PIL addition on the microcrystalline structure.
- Also, in the abstract and conclusions, it should be noted that the use of a PIL did not result in an improvement of the separation of CO2 from inert gases, which is the most important result of this paper and the general reason for the use of IL additives.
- On pag 2 line 74: “are virtually non-existent”. What does the authors mean with “virtually”? Please rephrase.
- The right reference for the Dual Mode Sorption model is J. Polym. Sci. 27 (1958) 177–197. Please use it instead of ref 4.
Reviewer 2 Report
In this manuscript, the PEVI-DEP/Pebax® 1657 composite membranes were fabricated and the effect of PEVI-DEP content on morphology of membranes and gas transport properties were investigated. The gas separation performance of composite membranes is not good due to the poor compatibility between PIL and Pebax® 1657. Simple blending is not a good way to improve the gas separation performance of membranes. Also, many problems exist in the manuscript. The authors need to improve the quality of this paper.
- The authors should reasonably explain why the incorporation of the PEVI-DEP improved the gas permeability of Pebax®1657 for H2, He, O2 and N2 but not CO2. And the authors attributed this phenomenon to the already high CO2 sorption capacity of Pebax®1657. This is unreasonable. As the authors stated in the introduction, the CO2 permeability of Pebax®1657 can be further enhanced by introducing ILs such as 1-n-alkyl-3-methylimidazolium cation.
- In the Abstract and Conclusions, the authors concluded that “the Pebax® and PEVI-DEP showed similar affinity towards CO2” (sorption tests with CO2 showed a similar gas uptake for Pebax ® 1657 and PEVI-DEP). However, there is no relevant CO2 adsorption data of the PEVI-DEP in the Results and Discussion section, which is recommended to supply.
- In the sentence “Importantly, PEVI-DEP (40wt.%) incorporation improved permeability and selectivity by ca. 100% for light gas separations, but higher PEVI-DEP concentrations lead to a decline in the transport properties”, the “ca. 100% for light gas separations” should be exact. “ca. 100%” ? what are the light gases?
- The graphs in this paper need to be modified and improved, e.g. the line and coordinate scale are not clear in Figure 3, 4, 5 and 6.
- For “3.1. Morphology and bulk properties”, more discussion about mechanical properties of neat and composite membranes are essential. Mechanical properties tests need to be supplemented in manuscript.
- Line 172, “We believe that PEVI-DEP clusters cause an increase of the distance between Pebax® 1657 polymeric chains” , too much clusters are observed in composite membranes. Does this cause interface defects? The clearer SEM images help to explain this suspicion.
- The XRD analysis of neat and composite membranes need to be supplemented in manuscript for investigating the chain distance in membrane.
- For “3.2. Gas Transport Properties”, “the CO2 and CH4 permeability remains almost constant or slightly decreases with PEVI-DEP content”, the authors attribute this to the high solubility of gas molecules. And “PEVI-DEP addition to the composites has an only marginal effect on the gas solubility”. The writing in this part is too confusing to understand. A clearer description and discussion is needed.
- Line 208-212, the writing needs to be improved, and clearer description and discussion can express ideas more clearly and help readers understand them.
- The CO2/CH4 selectivity of neat Pebax® 1657 and Pebax® 1657/PEVI-DEP membranes should be supplemented in manuscript Figure 3b, d and e.
Round 2
Reviewer 2 Report
Authors have revised carefully the manuscript based on the reviewers' comments. It is now suggested for publication on Membranes.